# The Important Role of the Radiologist in Determining the Indications for the Surgical Treatment of Neuroblastoma with Vascular Image-Defined Risk Factors: A Case Report

**DOI:** 10.3390/medicina57030279

**Published:** 2021-03-17

**Authors:** Patrycja Sosnowska-Sienkiewicz, Przemysław Mańkowski, Anna Wojas, Katarzyna Jończyk-Potoczna, Danuta Januszkiewicz-Lewandowska

**Affiliations:** 1Department of Pediatric Surgery, Traumatology and Urology, Poznan University of Medical Sciences, Szpitalna Street 27/33, 60-572 Poznan, Poland; mankowskip@ump.edu.pl; 2Department of Pediatric Radiology, Karol Jonscher Hospital, Szpitalna Street 27/33, 60-572 Poznan, Poland; a.d.grzybowska@gmail.com; 3Department of Pediatric Radiology, Poznan University of Medical Sciences, Szpitalna Street 27/33, 60-572 Poznan, Poland; potocznak@ump.edu.pl; 4Department of Pediatric Oncology, Hematology and Transplantology, Poznan University of Medical Sciences, Szpitalna Street 27/33, 60-572 Poznan, Poland; danuta.januszkiewicz@ump.edu.pl

**Keywords:** child, image-defined risk factors, neuroblastoma, radiology, surgery

## Abstract

The International Neuroblastoma Risk Group Staging System (INRGSS) is based on the age of patients and preoperative imaging, with attention paid to whether the primary tumor is affected by one or more of specific image-defined risk factors (IDRFs). This publication presents a 2.5-year-old boy with neuroblastoma who had an accidental ligation of the celiac trunk during tumor resection. The consequences of this complication were pancreatic and spleen ischemia and necrosis, ischemia, and perforation of the common bile duct, gallbladder, stomach, and duodenum. The aim of this publication was to highlight the great role of the radiologist in determining the indications for neuroblastoma tumor removal, especially with current vascular IDRFs, and to show how the radiologist’s insightful approach can save the patient from irreversible complications.

## 1. Introduction

Neuroblastoma accounts for approximately 8–10% of all childhood cancers and is the most common extra-cranial solid tumor in infants and young children [1,2,3]. About 60–70% of tumors are located in the abdomen, in the retroperitoneal space, in close proximity to large vessels. They can also occur in the neck, chest, pelvis, etc., [1,4,5]. The possibility of radical tumor removal is often impossible, and in addition, the location of the neuroblastoma may contribute to a number of complications in the course of the procedure. To minimize complications associated with the surgical procedure, the image-defined risk factors (IDRFs) were introduced in 2009. They have been designed to guide surgical management of neuroblastoma at diagnosis, in particular, to indicate whether a biopsy or attempted resection is recommended as the first surgical procedure. The same system of risk factors must be applied later during surgery after initial chemotherapy [6,7]. The IDRFs are de facto surgical risk factors but are identified by imaging, based on radiological criteria. The absence or presence of one or more of the image-defined risk factors, assessed at the moment of diagnosis, became the basis of the International Neuroblastoma Risk Group Staging System (INRGSS). The (INRGSS) [8,9], based on negative or positive IDRFs, respectively, classifies neuroblastoma into locoregional tumors—stages L1 or L2. Metastatic tumors are defined as stage M, except for stage MS, in which metastases are confirmed to the skin, liver, and/or bone marrow in children younger than 18 months of age [6,7].

We present a 2.5-year-old boy with neuroblastoma who had an accidental ligation of the celiac trunk during tumor resection. The consequences of this complication were pancreatic and spleen ischemia and necrosis, ischemia, and perforation of the common bile duct, gallbladder, stomach, and duodenum. The aim of this publication was to highlight the great role of the radiologist in determining the indications for neuroblastoma tumor removal, especially with current vascular IDRFs, and to show how the radiologist’s insightful approach can save the patient from irreversible complications.

## 2. Case Report

We present a retrospective analysis of the verification of radiological examinations performed on a 2.5-year-old boy with neuroblastoma. The boy was admitted to the hospital with a tumor in the abdominal cavity. Imaging examinations—ultrasonography (USG) and computed tomography (CT)—showed a solid mass, anteriorly from the right kidney and the thoracolumbar spine, ranging from Th11 to L3, with significant contrast enhancement. The lesion infiltrated the following organs: pancreas, the right adrenal gland, the liver’s visceral surface, the right kidney, the duodenum, the right side of the diaphragm, the right psoas muscle, and showing the neoplastic infiltration. The tumor compressed the inferior vena cava (VCI), which was about 2.5 cm away from the spine and was narrowed to 4 mm. Moreover, the tumor surrounded the following vessels: aorta (70–75% of the circumference), celiac trunk (100% of the circumference), common hepatic artery (100% of the circumference), superior mesenteric artery (100% of the circumference), VCI (50% of the circumference), right renal vein (50%), left renal vein (100%), splenic vein (50% of circumference), portal vein (50% of circumference). The dimensions of the lesion were: 4.9 cm × 4.6 cm × 7.3 cm (Figure 1a).

Computed tomography, magnetic resonance (MR), MIBG (metaiodobenzylguanidine) scintigraphy, trepanobiopsy, and bone marrow biopsy excluded distant metastases. According to the INRGSS, the tumor was classified as L2 (IDRFs present). Preoperative chemotherapy was started.

The histopathological examination revealed differentiated type neuroblastoma, partially mature, with medium density tumor tissue, with a low mitotic index and favorable histology (FH). In the immunohistochemical examination NSE (Neuron-specific enolase) (+), Synaptophysin (−), CD56 (+), S-100p (+), Ki-67 (+) were analyzed, and about 20% of the cells were positive. N-myc (−) gene amplification was absent.

The control CT, which was performed eight weeks from the moment of diagnosis, showed the reduction of the tumor volume. The mass was 35% smaller (Figure 1b). The decision about resection with further postoperative chemo- and/or radiotherapy was made.

Laparotomy and non-radical resection of the mass was performed. The tumor was dissected subtotally from surrounding organs and vessels. During operation, it was necessary to ligate a vessel with a diameter of 4 mm, entering centrally to the tumor. In addition, 48 h after the surgery, the boy presented abdominal pain. Ultrasonography did not visualize the flow through the splenic artery; the splenic vein was collapsed. There was flow through the common hepatic artery, portal vein, and other visualized vessels in the abdominal cavity. Urgent angiography did not reveal any contrast in the celiac trunk. In the next three months after the first operation, the child had to undergo a total of eight further surgical interventions related to ischemia of the abdominal organs.

These operations included the following:Laparotomy, partial spleen resection;Laparotomy, cholecystectomy, resection of the body, and tail of the pancreas;Laparotomy, resection of the remaining necrotic part of the spleen, and removal of the hematoma in the spleen area;Endoscopy, insertion of a stent into the common bile duct, and clip for duodenal perforation;Laparotomy, duodenal perforation suturing, pyloric stitching, common bile duct perforation suturing, replacement of the common bile duct stent with a T-drain, anastomosis between the first jejunal loop behind the Treitz ligament and the stomach, and drainage of the area where the T-drain was introduced into the bile duct;Laparotomy, duodenal perforation suturing, and insertion of a Nelaton 4 Fr catheter into the common bile duct;Laparotomy, pancreatic resection with duodenum (leaving a hook-shaped part), and insertion of a Nelaton 8 Fr catheter into the biliary tractLaparotomy, gastric sewing, and replacement of the Nelaton 8 Fr catheter for the biliary tract under X-ray control.

Such a dramatic complication after the neuroblastoma tumor removal procedure has not been observed in our center or in the available literature so far; therefore, radiologists were asked to re-analyze all CT images. Retrospectively performed 3D reconstruction of CT examinations allowed to precisely show the location of the tumor in relation to the large vessels in the abdominal cavity, including the abdominal trunk (Figure 2 and Figure 3). The results of CT examinations presented in this way made it possible to precisely locate the course of the vessels in relation to the tumor. Moreover, they showed vascular IDRFs present at the time of diagnosis and preoperatively. The results of computed tomography presented in this way helped us to re-analyze the previous examination without reconstruction. The listed elements were visible but less noticeable.

## 3. Discussion

Since the introduction of image-defined risk factors (IDRFs) to the International Neuroblastoma Risk Group Staging System (INRGSS) in 2009, their role in predicting surgical complications has been analyzed by many authors [8,9]. The aim of the image-defined risk factors (IDRFs) was a better definition of pretreatment risk factors. For example, abdominal and pelvic IDRFs include tumor-infiltrating porta hepatis or hepatoduodenal ligament, encasing branches of the superior mesenteric artery at mesenteric root, encasing origin of celiac axis or superior mesenteric artery, invading one or both renal pedicles, encasing aorta or vena cava, encasing iliac vessels, and pelvic tumor crossing sciatic notch [7]. 

The INRGSS classifies neuroblastoma into the following: L1—disease is localized and does not involve vital structures, is limited to one body compartment and L2—a localized disease with positive IDRFs, M- distant metastases present, and MS- metastatic disease confined to the skin, liver, and/or bone marrow in children <18 months of age) [1]. According to the INRGSS classification, the presented patient was in group L2. Three IDRFs have been identified (the tumor encased aorta and vena cava, the origin of the celiac axis, and branches of the superior mesenteric artery at the mesenteric root) (Figure 1). That is why two cycles of chemotherapy were initially applied, resulting in a volume reduction of over 35%. Most patients in the L1 stage can be treated using surgery only. Barak et al. mentioned that 80% of the tumors, including high-risk neuroblastoma tumors, had total gross resection of the tumor with minimal operative morbidity and no mortality. Overall, 88% of children had greater than 90% resection of their lesion, and the three-year survival was 84% [3]. These results confirmed the sense of surgical treatment in our patient. Unfortunately, the postoperative complication related to the ligation of the vessel, which was identified as the vessel supplying blood to the tumor, resulted in many serious complications. In the postoperative angiography, this vessel was identified as a coeliac trunk. The coeliac artery and its branches supply blood to the spleen, pancreas, liver, stomach, and part of the duodenum. The precise knowledge of the localization of the celiac trunk in context to the tumor represents an essential prerequisite for the successful removal of the tumor [4,5]. Our patient presented type 1 of coeliac trunk variation (Figure 4). 

In the literature, consequences of ligation or occlusion of the coeliac trunk are ischemia of the liver, gallbladder, and spleen [10,11]. It is possible that collateral visceral circulations may compensate for the occlusion or ligation of the celiac and superior mesenteric artery. In our patient, we observed all of the described complications associated with the ligation of the visceral trunk. The boy presented ischemia and necrosis of the common bile duct, gall bladder, pancreas, and spleen. Transient liver damage was also observed. In the available literature, no such severe complication has been described.

In the postoperative workup of this complication, 3D reconstructions were made in dedicated syngo.via imaging software for the TK Dual Source Somatom Force scanner (Siemens, Erlangen, Germany) (Figure 2 and Figure 3). The representation of the 3D reconstructions clearly showed the relationship of the tumor to the surrounding structures. They made operators aware that it was practically impossible to remove the entire lesion without damaging the coeliac trunk. The vessel entered centrally into the tumor (Figure 2). After re-examining the original study, surgeons found features so emphasized on the 3D reconstruction scans. After careful analysis, they were noticeable but much more difficult. We cannot say that 3D reconstructions are necessary every time because not every center has this kind of possibility, but they are very beneficial in preparing the surgeon for complicated operations.

## 4. Conclusions

Detailed imaging diagnostics (CT, MR, angiography), 3D reconstruction, and radiologist’s participation in the assessment of tumor resectability are crucial before deciding on the surgery in children with neuroblastoma and any solid tumor.

We hypothesize that demonstration of images and discussion of management in a multidisciplinary tumor board meeting increases the chances of each patient with a solid tumor for recovery and survival.

An initial and formal cancer committee meeting is the standard of care for all childhood cancers.

Retrospective analysis showed that 3D images are very helpful and able to emphasize the risk of complications in children operated on for solid tumors.

The presented case taught us that in every child with neuroblastoma, especially with vascular IDRFs present, CT studies must be analyzed carefully by the entire team. The surgeon before deciding on surgery should also have the radiologist’s opinion about the feasibility of tumor resection and the possible risk of surgical complications. A dedicated tumor board team consisting of a radiologist, surgeon, oncologist, and the rest of the team are responsible for the treatment of children with solid tumors, including neuroblastoma.

Surgery of patients with neuroblastoma and any solid tumor should be performed only in highly specialized centers with extensive experience in surgical treatment.

## Figures and Tables

**Figure 1 medicina-57-00279-f001:**
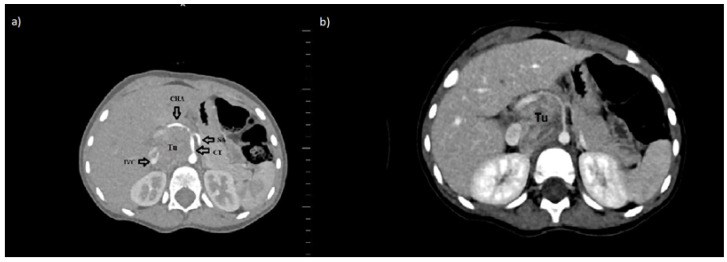
Axial abdominal computed tomography scan in venous phase showed a retroperitoneal tumor in our patient a diagnosis (**a**) and after two courses of chemotherapy with Vepeside and Carboplatin (**b**). Abbreviations: CT—coeliac trunk, CHA—common hepatic artery, SA—splenic artery, IVC—inferior vena cava, Tu-tumor.

**Figure 2 medicina-57-00279-f002:**
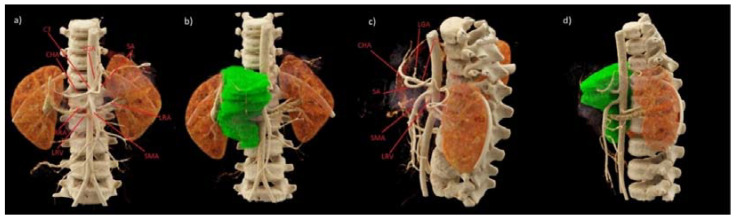
Preoperative abdominal computed tomography (CT) scan in the arterial phase with 3D VRT reconstructions (**a**–**d**), additionally visualizing the mass of the tumor (**b**,**d**). CT—coeliac trunk, SA—splenic artery, CHA—common hepatic artery, LGA—left gastric artery, SMA—superior mesenteric artery, RRA—right renal artery, LRA—left renal artery, LRV—left renal vein.

**Figure 3 medicina-57-00279-f003:**
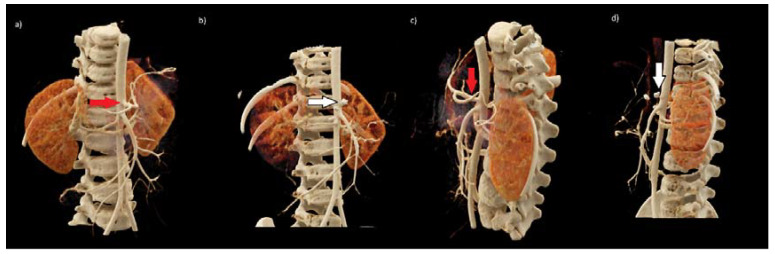
Preoperative (**a**,**c**) and postoperative (**b**,**d**) 3D VRT reconstructions of vessels in abdominal CT scan in the arterial phase. Red arrows (**a**,**c**)—normal variation of the coeliac trunk with its branches. White arrows (**b**,**d**) note absent depiction of the coeliac trunk and its main branches. In the typical location of the coeliac trunk, there are vascular clips; nearby next two clips are visible.

**Figure 4 medicina-57-00279-f004:**
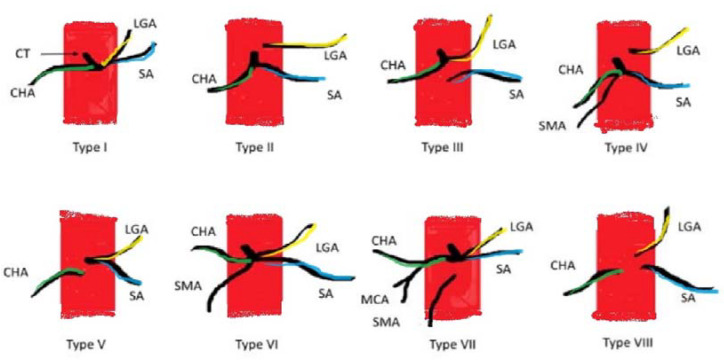
A scheme of types of coeliac trunk variations according to Uflacker’s classification. CT—coeliac trunk, CHA—common hepatic artery, LGA—left gastric artery, SA—splenic artery, SMA—superior mesenteric artery, MCA—middle colic artery.

## Data Availability

Data available on request due to restrictions.

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
