# Peer review of "The Important Role of the Radiologist in Determining the Indications for the Surgical Treatment of Neuroblastoma with Vascular Image-Defined Risk Factors: A Case Report"

_medicina, 2021, doi:10.3390/medicina57030279_

Round 1
Reviewer 1 Report
The manuscript entitled:" The key role of the radiologist in determining the indications for the surgical treatment of neuroblastoma with vascular
IDRFs. Case report." focused in a case report where neuroblastoma patient was administrated according to IDRFs factoirs is well writtenm and requires some minor revisions to be suitable for publication:
- please, could the authoirs provide more data about histological classification of neuroblastoma tumors? According to this aspect, please could the authors define histological report for this patient?
- Please, could the authors verify if other literature data reported the same role for radiological parameters in the managment of neuroblastoma patients?
- Please, could the authors report the if an update in IDRFs parameters shoulld be introduced to administrate patientys with difficult clinical condition?
Author Response
Dear Reviewer,
I am very grateful for the review of the article “The key role of the radiologist in determining the indications for the surgical treatment of neuroblastoma with vascular Image Defined Risk Factors. A case report.” I would like to address your comments and suggestions.
1.I am very grateful for that notice regarding histopathological examiantion.
Shimada developed the first histology grading system in 1984 to classify neuroblastic tumors by histologic features. Features including degree of stroma present, grade of differentiation, mitosis-karyorrhexis index (MKI), presence of nodules, and age were utilized to define groups with either favorable or unfavorable prognosis.
The International Neuroblastoma Pathology Classification (INPC) was then developed in 1999 to update these histologic factors impacting prognosis. INPC incorporates multiple factors consisting of diagnostic category (accounting for quantity of Schwannian stromal development and grade of tumor differentiation), MKI, and age to ultimately define tumor histology as favorable versus unfavorable. Diagnostic categories include ganglioneuroma (Schwannian stroma-dominant) with mature or maturing subtypes; ganglioneuroblastoma, intermixed (Schwannian stroma-rich); ganglioneuroblastoma, nodular (composite Schwannian stroma-rich/stroma-dominant and stroma-poor); and neuroblastoma (Schwannian stroma-poor) with undifferentiated, poorly differentiated, and differentiating subtypes. This provides greater subdivision than the Shimada classification that only defined stroma-poor and stroma-rich. Grade was further subdivided as well to include three categories (differentiating, poorly differentiated, and undifferentiated) instead of the two included in the Shimada system (differentiating and undifferentiated). MKI reflects the degree of cell replication seen in a high-power microscope field and categorized as low (<2% or <100/5000 mitotic and karyorrhectic cells), intermediate (<2%–4% or <100–200/5000 mitotic and karyorrhectic cells), and high (>4% or >200/5000 mitotic and karyorrhectic cells). Age is not incorporated into prognostic grouping for ganglioneuroblastoma intermixed and ganglioneuroma, which fall within the favorable histology group or ganglioneuroblastoma, nodular which is considered an unfavorable histology. Age and MKI, however, impact prognostic grouping within tumors categorized as neuroblastoma. Neuroblastoma tumors with favorable histology follow a framework of age-linked maturation and include poorly differentiated (age <1.5 years) to differentiating (age <5 years) neuroblastoma and should have low (age <5 years) or up to intermediate (age <1.5 years) MKI. In contrast, tumors with unfavorable histology demonstrate features that suggest aggressive growth and have immature histologies for the age of the patient. Within neuroblastoma tumors, the unfavorable histology group includes undifferentiated histology (in any age) or poorly differentiated subtype (age ≥1.5 years) or any subtype (age ≥5 years). Further, those with high MKI (in any age) or intermediate (age ≥1.5 years) qualify as having unfavorable histology. Age is included in INPC and contributes to the prognostic significance of the histologic groups. Thus, within the risk classification system, age is weighted twice—both within INPC and independently. (Sokol E, Desai AV. The Evolution of Risk Classification for Neuroblastoma. Children (Basel). 2019;6(2):27. Published 2019 Feb 11. doi:10.3390/children6020027)
Histopathological classification of neuroblastoma in our patient was included in the publication.
The histopathological examination was diagnosed with differentiating type neuroblastoma, partially maturing, with medium density tumor tissue, with a low mitotic index and favorable histology (FH). In the immunohistochemical examination NSE (+), Synaptophysin (-), CD56 (+), S-100p (+), Ki-67 (+) were analyzed, and about 20% of the cells were positive. N-myc (-) gene amplification was absent. (line 75-79)
- I am very grateful for that notice. The role of the IDRFsin predicting has been analyzed during many studies. Information about these was included in the additional citation [9].
- Parhar D, Joharifard S, Lo AC, Sclosser Mary-Pat, Daodu OO. How well do image-defined risk factors (IDRFs) predict surgical outcomes and survival in patients with neuroblastoma? A systematic review and meta-analysis. Pediatric Surgery International. 2020;36:897-907.
- Thank you very much for this question. Based on the performed study, we do not think that it is necessary to update IDRFs. We would like to emphasize that detailed imaging diagnostics (CT, MR, angiography), 3D reconstruction, and radiologist's participation in tumor resectability assessment are necessary before deciding on the surgery in children with neuroblastoma with IDRFs.
This conclusion was included as one of many at the end of the publication.
Detailed imaging diagnostics (CT, MR, angiography), 3D reconstruction, and radiologist's participation in tumor resectability assessment are necessary before deciding on the surgery in children with neuroblastoma with IDRFs. (line 184-186)
Once again, thank you for your review and valuable comments.
Yours faithfully,
Patrycja Sosnowska-Sienkiewicz, MD, PhD

Reviewer 2 Report
The authors report one of the most serious complications that neuroblastoma resection can bring. The authors are to be thanked for writing about the complication in this openness with the intention of preventing similar courses in other patients. But unfortunately, they draw the wrong conclusions from what has happened. That the truncus was surrounded by the tumor can be seen very clearly in the initial imaging. Therefore, during resection, the truncus should have been clearly identified and preserved. This is a procedure that is necessary in a large number of retroperitoneal NB and is practically part of the surgery.
Several indications suggest that the authors have little to no experience in NB surgery. For example, a close look at the relationship of the NB to the arteries in the upper abdomen in cross-sectional imaging before surgery is part of adequate preparation. Whether this is called IDRFs or not is beside the point. Furthermore, the NB is never fed by arteries with a diameter of 4 mm, because the masses in the area of the truncus are lymph node metastases, which have a different blood supply. Accordingly, such thick arteries are indeed pulling into the tumor, but they also come out at the other end and have to be dissected over their entire length. It is true that collateral formation occurs when the blood supply via the truncus fails, but unfortunately only in the case of a very slowly progressing occlusion. A transection almost always ends in the described complications. If the liver does not still have an accessory blood supply from the SMA, it can unfortunately also be assumed that the child will develop cirrhosis of the liver and bile duct stenosis in the next few years.That the surgeon must be guided by the IDRFs prior to surgery is correct. However, the IDRFs were unambiguous. Therefore, it cannot be argued that the incident could have been avoided by an even better presentation of the IDRFs. The conclusion that such patients are inoperable is wrong and dangerous, because this could lead to an inadequate treatment of future children not getting this necessary operation and getting an inadequate therapy with mostly fatal end. NB resection is one of the most complex operations in childhood. Accordingly, it is necessary to demand that the operation is performed in a center that can demonstrate the best possible expertise in the treatment of NB.
The authors are to be highly commended for reporting the complication so openly. This is important and right to prevent harm to future patients. But unfortunately they draw the wrong conclusions, so that the statement made may lead to wrong conclusions for future patients.
Author Response
Dear Reviewer,
I am very grateful for the review of the article "The key role of the radiologist in determining the indications for the surgical treatment of neuroblastoma with vascular Image Defined Risk Factors. A case report." I would like to address your comments and suggestions.
On the beginning, I would like to thank you for the appropriate summary of our publication.
Yes, our main aim was to honestly admit to the consequences of ligation of the celiac trunk. We wanted to warn all surgeons who operate on patients with neuroblastoma, especially those with IDRFs present.
In our example, you can see how such a decision exposed the patient to irreversible consequences and how much stress it was for the surgeon.
The initial examination showed the anatomical relation of the tumor to the surrounding structures. We were aware of the severity of the procedure performed. As you can see, not entirely. Unfortunately, the change of anatomical relations during the operation was a complete surprise for us. It is possible that the lack of experience in such difficult procedures was the reason for this. However, we are not trying to explain our mistakes. We want to warn fellow surgeons of what consequences the operation of neuroblastoma, especially with IDRFs, may lead to. Also, we want to emphasize the role of cooperation with a team of radiologists and the fact that maybe if we had 3D reconstructions before surgery, they would protect the child and us from such severe ligation effects on the trunk and stress.
We want to emphasize that we should protect ourselves against this type of treatment in all possible ways.
We added a request appropriate to this comment, for which we would like to thank you.
Surgery of patients with neuroblastoma should be performed only in highly specialized centers with extensive surgical treatment experience. (line 201-202)
We want to leave unchanged in our publication to underline the importance of IDRFs, but only as underlined below. We hope it can be accepted.
The presented case taught us that in every child with neuroblastoma, especially with vascular IDRFs present, CT studies must be analyzed carefully by the radiology team. (line 194-195)
Moreover, on the beginning of the discussion, the role of IDRFs was supported by additional citation [9].
Once again, thank you for your review and valuable comments.
Yours faithfully,
Patrycja Sosnowska-Sienkiewicz, MD, PhD
Round 2
Reviewer 2 Report
Dear Authors,
Unfortunately, you have not managed to address the main points of criticism of the work. Of course, the treatment of such complex tumors must be carried out in an interdisciplinary team, but this is standard practice. However, it is not standard to demand 3D reconstruction. That is a nice to have, but by no means elementary. The statement that the radiologist should examine the CT in the case of IDRFs is disturbing. And not in the case of no IDRFs? The discussion and summary is inconclusive in its reasoning, which negates the scientific value of the paper.
Author Response
Dear Reviewer,
I am once again very grateful for the review of the article “The key role of the radiologist in determining the indications for the surgical treatment of neuroblastoma with vascular Image Defined Risk Factors. A case report.”
Of course, we completely understand and respect your advice.
Perhaps our approach may differ because we look at the same problem but from the perspective of different medical specialties.
We tried to improve the work so that it would be attractive to everyone.
- We changed the title of the publication to “The important role of the radiologist in determining the indications for the surgical treatment of neuroblastoma with vascular Image Defined Risk Factors. A case report.”
- We changed the results and added a sentence, which summarizes the results and suggests that reconstruction is not necessary but helpful.
The results of computed tomography presented in this way helped us to re-analyze the previous examination without reconstruction. The listed elements were visible, but less noticeable. (line 121-123)
- We changed the final part of the discussion.
We delated the sentence “The performed vascular reconstructions changed the surgeon's perspective on the operability of the lesion.”
We added the summary:
After re-examining the original study, surgeons found features so emphasized on the 3D reconstruction scans. After careful analysis, they were noticeable but much more difficult. We cannot say that 3D reconstructions are necessary every time because not every center has this kind of possibility, but they are very beneficial in preparing the surgeon for complicated operations. (line 180-184)
- We changed our conclusions completely.
Detailed imaging diagnostics (CT, MR, angiography), 3D reconstruction and radiologist's participation in the assessment of tumor resectability are crucial before deciding on the surgery in children with neuroblastoma and any solid tumor.
Cooperation in a multidisciplinary team increases the chances of each patient with a solid tumor for recovery and survival.
An initial and formal cancer committee meeting is the standard of care for all childhood cancers.
Retrospective analysis showed that 3D images are very helpful and able to emphasize the risk of complications in children operated on for solid tumors.
The presented case taught us that in every child with neuroblastoma, especially with vascular IDRFs present, CT studies must be analyzed carefully by the entire team. The surgeon before deciding on surgery should also have the radiologist's opinion about the feasibility of tumor resection and the possible risk of surgical complications. A dedicated tumor board team consisting of a radiologist, surgeon, oncologist and the rest of the team are responsible for the treatment of children with solid tumors, including neuroblastoma.
Surgery of patients with neuroblastoma and any solid tumor should be performed only in highly specialized centers with extensive experience in surgical treatment.
(line 187- 204).
All corrections are highlighted in yellow in the text.
Once again, thank you for your review and valuable comments.
Yours faithfully,
Patrycja Sosnowska-Sienkiewicz, MD, PhD
